# GEM: A GYM FOR AGENTIC LLMS

**Zichen Liu**[†1 2], **Anya Sims**[†1 3*], **Keyu Duan**[†1 2*], **Changyu Chen**[†1 4*], **Simon Yu**[6 9], **Xiangxin Zhou**[1*]
**Haotian Xu**[7], **Shaopan Xiong**[8], **Bo Liu**[2], **Chenmien Tan**[9], **Chuen Yang Beh**[2], **Weixun Wang**[8]
**Hao Zhu**[5], **Weiyan Shi**[6], **Diyi Yang**[5], **Michael Shieh**[2 10], **Yee Whye Teh**[3], **Wee Sun Lee**[2], **Min Lin**[1]
[1]Sea AI Lab [2]NUS [3]Oxford [4]SMU [5]Stanford [6]Northeastern [7]OpenRLHF [8]ROLL [9]RL2 [10]absolute AI

## ABSTRACT

The training paradigm for large language models (LLMs) is moving from static datasets to experience-based learning, where agents acquire skills via interacting with complex environments. To facilitate this transition we introduce GEM (General Experience Maker), an open-source environment simulator designed for the age of LLMs. Analogous to OpenAI-Gym for traditional reinforcement learning (RL), GEM provides a standardized framework for the environment-agent interface, including asynchronous vectorized execution for high throughput, and flexible wrappers for easy extensibility. GEM also features a diverse suite of environments, robust integrated tools, and single-file example scripts demonstrating using GEM with five popular RL training frameworks. Along with this, we also provide a set of baselines across 24 environments using REINFORCE with *Return Batch Normalization* (ReBN), which—unlike GRPO—is compatible with the full RL setting of dense per-turn rewards and offers better credit assignment. We further conduct apple-to-apple benchmarking of PPO, GRPO and REINFORCE in both single- and multi-turn settings using GEM to shed light on the algorithmic designs. Lastly, GEM also functions as a convenient evaluation toolkit besides a training environment. We hope this framework can help accelerate future agentic LLM research[1].

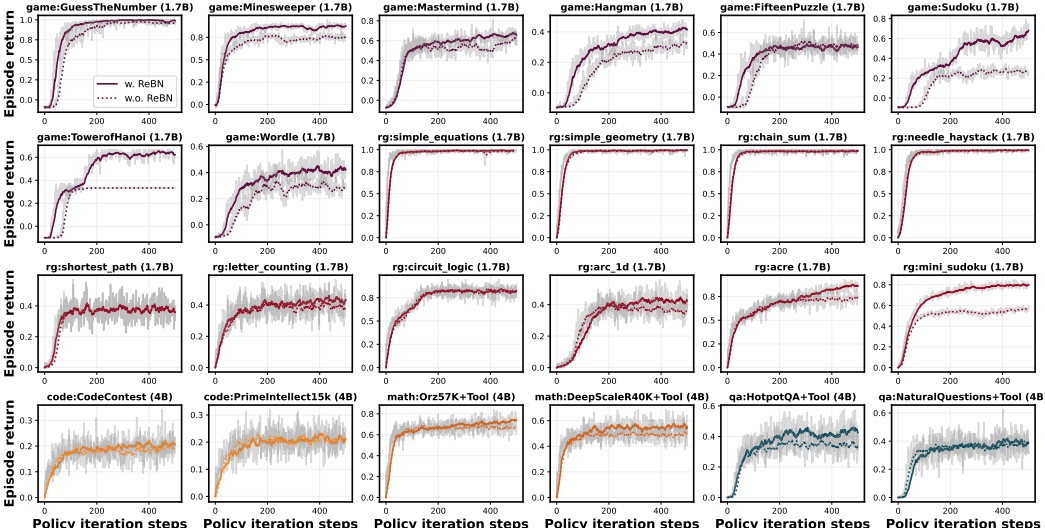

Figure 1: Learning curves of Qwen3-based agents across diverse environments of 5 categories: **game** (language games); **rg** (ReasoningGym); **code** (coding tasks); **math** (python-integrated math questions); **qa** (search-integrated general questions). All agents are learned via a simple yet general multi-turn algorithm based on REINFORCE (Algorithm 1). The comparison between two curves in each subplot illustrate the effectiveness of Return Batch Normalization (ReBN).

## 1 INTRODUCTION

Reinforcement learning (RL) (Sutton and Barto, 2018) has emerged as a powerful paradigm for improving the reasoning capabilities of large language models (LLMs) (OpenAI, 2024; Guo et al., 2025). By collecting *experience* in interactive environments, RL allows agents to learn complex,

---

[†]Equal contribution with random order. [*]Work done during their associate membership at Sea AI Lab.
[1]Code is available at: https://github.com/axon-rl/gem.

open-ended tasks without direct supervision (Silver and Sutton, 2025). This approach promises to create powerful agents for a variety of domains. For instance, an agent could develop entire software modules by writing, testing, and debugging code, while also adapting to integration failures or evolving requirements. Similarly, in scientific discovery, an agent could be trained to develop hypotheses, design relevant experiments, and adjust its long-term strategy based on the results.

However, current research on RL for LLMs has largely focused on single-turn tasks, such as answering math questions or retrieving specific data (Lambert et al., 2024; Guo et al., 2025). While these tasks are a valuable starting point, they significantly oversimplify multi-turn interactions (Liu et al., 2025a). This oversimplification means that algorithms which excel in the single-turn setting (e.g., GRPO (Shao et al., 2024)) are fundamentally inapplicable to full multi-turn problems. If the goal is to train agentic LLMs capable of long-horizon planning, trial-and-error, iterative refinement etc, it is crucial to transition to testbeds that support these more complex multi-turn interactions.

To facilitate this next step, we introduce GEM (General Experience Maker), an open-source environment framework for diverse, multi-turn, long-horizon tasks. Motivated by OpenAI-Gym (Brockman et al., 2016) which catalyzed research in traditional RL by providing a unified interface and standardized environments, GEM aims to provide analogous foundational infrastructure for LLM agents. GEM offers a diverse suite of environments spanning single- and multi-turn (over 100 turns) tasks (including tool integrated responses, reasoning games etc), flexible observation and action wrappers, asynchronous parallel execution, and a rich set of tools (python, search, and external MCP compatible tools). Additionally, GEM includes validated baselines and single-file training scripts showcasing seamless integration with five popular RL training frameworks (Oat, Verl, OpenRLHF, ROLL, and RL2—see Section 4.5).

Besides introducing the GEM framework, this paper also presents and discusses a simple yet effective algorithmic variant of REINFORCE (Williams, 1992) which incorporates *Return Batch Normalization* (ReBN), a useful technique similar to advantage normalization (Andrychowicz et al., 2021; Liu et al., 2025b) that brings consistent improvements (Figure 1). Unlike GRPO and its variants, REINFORCE with ReBN is fully compatible with the multi-turn RL setting, including turn-level dense rewards and arbitrary discount factors. We further compare REINFORCE-based algorithms with multi-turn PPO (Schulman et al., 2017) and GRPO, showing its theoretical connections and empirical tradeoffs. We also provide case studies on the impact of the discount factor $\gamma$ on multi-turn learning, extensive results of tool-integrated RL, and performance benchmarks on terminal and MCP usage of strong LLMs using GEM as a unified evaluation toolkit. We hope this framework will accelerate RL research on agentic LLMs and advance progress toward more capable and autonomous AI systems.

## 2 GEM ENVIRONMENTS

This section introduces GEM's core functionality, covering its main interface (Section 2.1), the environment design (Section 2.2), and advanced features such as asynchronous vectorization and modular wrappers (Sections 2.3 and 2.4).

### 2.1 INTERFACE

GEM employs a standardized environment interface closely following the well-established OpenAI Gym API with the main functions being `reset()` and `step()`. A basic agent-environment interaction loop is as follows (multi-agent interface shown in Appendix D.3):

```python
import gem
# gem.print_envs() # to list all available environments
env = gem.make("game:GuessTheNumber-v0")
observation, info = env.reset()

while True:
    # (1) Agent acting:
    action = env.sample_random_action()
    # action = agent.act(observation) # real acting by LLM sampling

    # (2) Environment execution:
    next_obs, reward, terminated, truncated, info = env.step(action)

    # (3) Agent learning:
    # agent.learn(observation, action, reward)

    observation = next_obs
    if terminated or truncated: break
```

## 2.2 TASKS AND TOOLS

GEM's core environment components are **tasks** and **tools**. Each combination of a task and an optional set of tools constitutes an environment that tests complex capabilities such as reasoning, multi-step planning, and tool use. These environments can therefore be used to benchmark LLMs and to test and develop new algorithms. GEM currently features seven main categories of tasks:

> **Math:** Solve math problems with chain-of-thought reasoning.
>
> **Math with image:** Solve geometry math problems with images using chain-of-thought reasoning.
>
> **Code:** Generate code to solve competitive programming problems.
>
> **Game:** Multi-turn text-based games adapted from TextArena (Guertler et al., 2025).
>
> **QA:** General, potentially knowledge-intensive questions (useful for testing search tool capability).
>
> **ReasoningGym:** A unified interface of ReasoningGym (Stojanovski et al., 2025) which provides 100+ single-turn verifiable tasks.
>
> **Terminal**: Perform complex tasks through a containerized terminal environment.

GEM's modular design simplifies task integration. Math (with images), code, and QA tasks can be integrated by simply providing a new dataset. Terminal tasks require a new Docker file, instructions, and test cases. New games and other custom tasks can be added by inheriting from GEM's environment base class and defining their state transition and reward logic. In addition, tasks can be augmented with any combination of tools. GEM currently supports:

> **Python:** Parses and executes code blocks, returning the stdout or execution error.
>
> **Search:** Parses a query, executes a search against an external engine, and returns the results.
>
> **MCP:** General tool calling to any external servers that conform to the model context protocol.

The use of tools converts single-turn tasks, like Math or ReasoningGym, into multi-turn tasks in which an agent can learn to call tools and adapt based on their output.

## 2.3 ASYNCHRONOUS VECTORIZATION AND AUTORESET

To facilitate efficient agent RL training, we support parallel execution of vectorized environments via asynchronous tool calls to collect episodes in batches. In addition to the latency reduction, the use of vectorized environments with autoreset streamlines the experience collection logic. Users can run a single `.reset()` at the initialization stage and simply continue with `.step()` in the following agent-environment loop for continuous data generation. In addition, the user code can use the returned `terminated` flag to prevent value bootstrapping across episode boundaries, ensuring the correctness of critic learning. An illustration of the autoreset mechanism can be found in Figure 2.

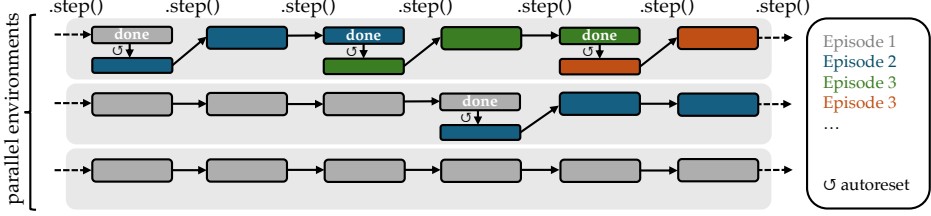

Figure 2: Illustration of autoreset in vectorized environments. Autoresetting resets the environment automatically after termination, allowing users to collect batches of episodes by simply running `.step()` without needing more complicated logic such as keeping track of whether individual episodes have terminated.

## 2.4 WRAPPERS

Like in OpenAI-Gym, GEM uses wrappers for easy extensibility. Observation wrappers, for example, control how the episode is converted into an observation. Options include observing just the most recent environment output, a concatenation of all previous environment outputs, a concatenation of all previous environment outputs and actions, or some parsed/summarized version of this. The

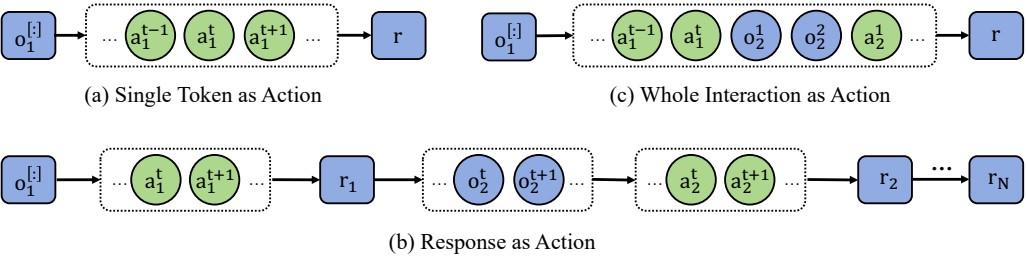

Figure 3: The illustration of different view of agentic RL. Green nodes denote tokens responsible for loss.

Python interpreter or database/web search tools are also formulated as wrappers which can be added on top of any specified task environment.

## 3   REINFORCEMENT LEARNING WITH GEM

In this section, we begin by describing the main RL formulations for LLMs, including their respective flexibilities and limitations (Section 3.1). Motivated by this, we then present our baseline algorithm which is applicable to the more flexible RL formulation (Section 3.2).

### 3.1   PRELIMINARY: LLMS AS AGENTS

There are three main ways of treating LLM-environment interactions in RL algorithms which each have different limitations and strengths:

**Action = Single token (Figure 3(a)):** The first approach is to treat each token generated by the LLM as an individual action (Ziegler et al., 2019). This, however, means that episodes are typically very long (thousands of tokens), and it also requires specifying the reward for the addition of every token, which is difficult to evaluate. Successful applications of RL in this formulation tend to use sparse outcome reward with discount factor $\gamma = 1$ (Guo et al., 2025).

**Action = Response (Figure 3(b)):** To avoid these complications the second approach is to treat a whole response (a sequence of tokens until an EOS) as a single action[2] (Ahmadian et al., 2024; Liu et al., 2025a). In answering math problems for example—currently the most common testbed for RL for LLMs—each episode contains a question and response. With this view all episodes therefore have length 1 and the RL problem essentially degenerates to contextual bandits (Abe et al., 2003). This is convenient as it means sample-based advantage estimation methods such as GRPO (Shao et al., 2024) can be applied efficiently, and these have been demonstrated to be highly effective. Extending to multi-turn episodes (e.g. for games or tool use), however, results in an issue: Multi-turn interactions have episode lengths $> 1$, meaning sample-based advantage estimation methods (e.g., Kazemnejad et al. (2025)) become infeasible (since they require collecting multiple episode completions from each turn (state) in the episode, leading to exponential complexity).

**Action = Whole interaction (Figure 3(c)):** One approach to make GRPO applicable to multi-turn interactions is to treat the whole interaction as a single action while masking the loss on tool outputs. This view again degenerates the full RL problem back to one-step RL or contextual bandits, meaning GRPO etc. can be applied. However, it requires two compromises: Firstly, it effectively fixes the discount factor at $\gamma = 1$, thus removing the incentive to solve problems quickly. This is significant, for example in Section 4.2 where we show how the optimal search algorithm is only recovered when setting $\gamma < 1$. Secondly, this approach is limited to single trajectory-level rewards, losing fine-grained per-turn credit assignment.

Many prior works make these concessions and use GRPO in multi-turn LLM RL (Cao et al., 2025; Jiang et al., 2025; Chen et al., 2025a; Jin et al., 2025; Feng et al., 2025a). However, to develop an algorithm compatible with the full RL setting, we go back to the second view (action=response) and employ a simple variant of REINFORCE with *Return Batch Normalization* (ReBN). Unlike GRPO, this algorithm is compatible with per-step dense rewards and arbitrary discount factors ($\gamma \leq 1$), thus making it significantly more flexible for optimizing LLMs in complex, multi-turn interactive settings.

[2]Ignoring token-level PPO clipping which has no effect if the updates are on-policy.

## 3.2 BASELINE ALGORITHMS

We start from the foundational on-policy[3] policy-gradient method REINFORCE (Williams, 1992), which optimizes the following objective:

$$\mathcal{J}_{\text{REINFORCE}}(\theta) = \frac{1}{N} \sum_{n=1}^{N} \sum_{t=0}^{T^{(n)}-1} G_t^{(n)} \log \pi_\theta(a_t^{(n)} | s_t^{(n)}), \tag{1}$$

where $N$ is the batch size, $[s_0, a_0, s_1, ..., a_{T-1}]$ is a sequence of states and actions making up a trajectory in which each $s_t$ and $a_t$ is itself a sequence of tokens, and $G_t = \sum_{k=t}^{T-1} \gamma^{k-t} r_k$ is the return. Though initially designed for single-turn problems (i.e., $T^{(n)} = 1$), GRPO can be extended to multi-turn tasks by sampling a group of $M$ trajectories per initial state and normalizing the trajectory-level reward for each group[4]:

$$\mathcal{J}_{\text{GRPO}}(\theta) = \frac{1}{N} \sum_{n=1}^{N} \frac{1}{M} \sum_{m=1}^{M} A_{\text{GRPO}}^{(n,m)} \sum_{t=0}^{T^{(n,m)}-1} \log \pi_\theta(a_t^{(n,m)} | s_t^{(n,m)}), \tag{2}$$

where $A_{\text{GRPO}}^{(n,m)} = (\sum_{t=0}^{T-1} r_t^{(n,m)} - \text{mean}(\mathbf{R}))/\text{std}(\mathbf{R})$ with $\mathbf{R} = \{\sum_{t=0}^{T-1} r_t^{(n,m)}\}_{m \in [1,...,M]}$. However, this approach has poor credit assignment for multi-turn problems because all turns in the trajectory share the same advantage estimation, and improving it typically requires tree-like sampling which leads to combinatorial explosion. To bypass the expensive sampling from each turn, we can learn a value function to estimate the return $G_t$, known as *critic* (Sutton and Barto, 2018), which in turn guides the policy learning in the actor-critic architecture. We can compute GAE (Schulman et al., 2015) for the advantage actor-critic (A2C) objective:

$$\mathcal{J}_{\text{A2C}}(\theta) = \frac{1}{N} \sum_{n=1}^{N} \sum_{t=0}^{T^{(n)}-1} A_{\text{GAE},t}^{(n)} \log \pi_\theta(a_t^{(n)} | s_t^{(n)}). \tag{3}$$

To retain the benefits of fine-grained and stable advantage estimation without the combinatorial explosion or learning an additional critic, we instead use *Return Batch Normalization* (ReBN). For ReBN the per-transition returns $G_i$ are normalized over the *whole batch of transitions*:

$$\mathcal{J}_{\text{REINFORCE+ReBN}}(\theta) = \frac{1}{N} \sum_{n=1}^{N} \sum_{t=0}^{T^{(n)}-1} A_{\text{ReBN},t}^{(n)} \log \pi_\theta(a_t^{(n)} | s_t^{(n)}), \tag{4}$$

where $A_{\text{ReBN},t}^{(n)} = (G_t^{(n)} - \text{mean}(\mathbf{G}))/\text{std}(\mathbf{G})$, with $\mathbf{G} = \{G_t^{(n)}\}_{n \in [1,...,N], t \in [1,...,T^{(n)}-1]}$. Each of these algorithms trains the agent by iterating between two main phases: (A) data collection and (B) policy update. We present the RL loop of Equation (4) in Algorithm 1 in Appendix C due to space constraint.

## 4 EMPIRICAL STUDIES WITH GEM

In this section, we demonstrate how GEM can facilitate RL research on agentic LLMs through a series of empirical studies. These include a comprehensive apples-to-apples algorithm benchmarking across eight GEM environments (Section 4.1); analyses of the effects of the discount factor $\gamma$ and tool integration (Sections 4.2 and 4.3); an examination of cross-task generalization (Section 4.4); and, finally, a demonstration of GEM's compatibility with five RL training frameworks along with their easily accessible infrastructure benefits (Section 4.5). RL results in a vision-language environment and analysis of a multi-agent environment can be found in Appendices D.2 and D.3.

## 4.1 BENCHMARKING RL ALGORITHMS FOR LLMS

Benchmarking has been critical for the progress of RL, with OpenAI-Gym providing standardized environments that enabled systematic evaluation of algorithms (Raffin et al., 2021; Huang et al., 2022; 2024). Following this paradigm, GEM offers a unified testbed for agentic LLMs, where prior work often relied on bespoke tasks that complicate fair comparison. We benchmark all baseline algorithms introduced in Section 3.2 (GRPO, PPO[5], REINFORCE, ReBN) across eight GEM environments under a unified experimental protocol. All algorithms are implemented using Oat (Liu et al., 2024) with hyperparameters detailed in Appendix F. Results are evaluated by mean episode return, sample efficiency, and stability.

---

[3] Orthogonally, we can also utilize proximal updates (Schulman et al., 2017) to improve sample efficiency.

[4] This is not the original GRPO because we fixed the length bias as noted by Liu et al. (2025b).

[5] PPO in this work generally refers to *turn-level PPO* instead of token-level PPO commonly seen in single-turn dialogue scenarios (Ouyang et al., 2022).

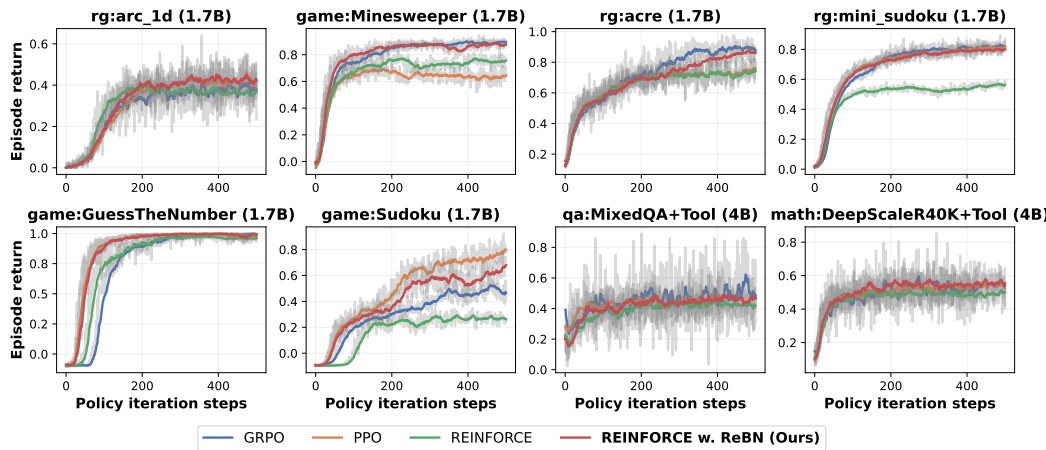

Figure 4: Algorithm benchmarking using eight representative environments from GEM. All agents are trained from `Qwen3-{scale}-Base` models, with `scale` specified in each plot. **rg** refers to single-turn reasoning tasks from ReasoningGym; **game** consists of long-horizon language games; **qa** and **math** are tool-integrated multi-turn environments.

We present all learning curves in Figure 4. We first observe that in all three single-turn environments (labeled with **rg**), GRPO performs reasonably well, defending its effectiveness in single-step RL with verifiable rewards. However, GRPO falls short when it comes to multi-turn environments (`GuessTheNumber` and `Sudoku`), where dense per-turn rewards are available and more fine-grained credit assignment is necessary for efficient policy learning, due to a constant advantage estimation across all steps. Such effects are the most profound when the environment's reward structure is inherently non-sparse (**qa** and **math** is less so).

In contrast to GRPO, REINFORCE and PPO are natively suitable for multi-turn RL. We find that vanilla REINFORCE is readily a strong baseline in most environments, but it might suffer from suboptimal convergence (e.g., two `Sudoku` environments). We hypothesize that this might be because the raw return calculation of vanilla REINFORCE can be sensitive to reward shaping, thus hindering exploration; we defer an in-depth ablation study to Appendix D.1. On the other hand, PPO is generally performant, attaining the best episode return in the complex and long-horizon `Sudoku` environment. This performance advantage can be attributed to a well-learned critic, but it is also deemed difficult to robustly learn an accurate critic (Van Hasselt et al., 2018; Kazemnejad et al., 2025) (as evidenced by the poor performance of PPO in `Minesweeper`), inviting future works to go in this direction.

Finally, we investigate the proposed REINFORCE variant, which incorporates a simple Return Batch Normalization (ReBN) technique. Results in both Figures 1 and 4 show that ReBN consistently improves on vanilla REINFORCE by a large margin, suggesting the empirical benefits of adaptive normalization of policy gradient coefficients. Moreover, ReBN outperforms or is comparable with PPO and GRPO in all evaluated environments, rendering it the strongest baseline without expensive computations, such as critic learning or extensive rollouts.

## 4.2 DISCOUNT FACTOR $\gamma$ MATTERS

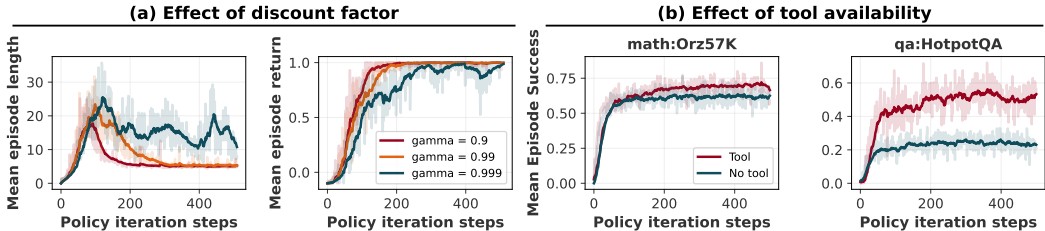

Figure 5: **(a)** Average number of turns and episode return when trained with different discount factors. **(b)** Comparative experiment results on tool availability.

Next, we investigate the effect of the discount factor $\gamma$. A key motivation for REINFORCE+ReBN over GRPO is its compatibility with arbitrary discount factors. To investigate the effect of this

we trained the `Qwen3-1.7B-Base` model (Yang et al., 2025) using REINFORCE+ReBN on the `GuessTheNumber` environment. In this environment the agent must guess a hidden number randomly selected between 1 and 50. At each turn the agent may guess, and receives feedback as to whether the hidden number is larger or smaller. The optimal strategy is therefore *binary search*.

As shown in Figure 5(a), as expected, smaller $\gamma$ values naturally encourage solutions with fewer turns and drive convergence to the optimal turn count ($\log_2(50) \approx 5.6$)—achievable only through binary search. Example interactions are included in Appendix B. As discussed in Section 3.2, the natural efficiency incentive from $\gamma < 1$ is not compatible with GRPO. Instead, prior works using GRPO hyperparameter tune the environment's maximum number of turns to get efficient agent behavior (Xue et al., 2025).

### 4.3 TOOL-INTEGRATION IN MATH AND QUESTION-ANSWERING TASKS

GEM is designed with modular support for external tools, enabling seamless integration into a range of tasks. To empirically assess the impact of tool use, we focus on two domains: Math and Question-Answering (QA).

Table 1: Math benchmark scores for four agents, evaluated with and without tool access and RL training. Note: scores should be interpreted relative to other values here due to sensitivity to the grader code (see Section 4.3).

| Qwen3-4B-Base | Base (no tool) | Base (with tool) | Base + RL (no tool) | Base + RL (with tool) |
|---|---|---|---|---|
| AIME24 | 10.0 | 6.7 | 16.7 | 30.0 |
| AMC | 39.8 | 50.6 | 49.4 | 67.5 |
| MATH500 | 61.0 | 62.4 | 67.4 | 71.0 |
| MinervaMath | 36.4 | 30.1 | 40.1 | 40.4 |
| OlympiadBench | 29.5 | 31.0 | 33.5 | 39.9 |
| Average | 35.3 | 36.2 | 41.4 | **49.8** |

We first investigate the effect of GEM's Python tool on Math tasks. Starting from the base model `Qwen3-4B-Base`, we finetune on the `math:Orz57K` environment, training two variants: one with Python tool integration and one without. The base model and both finetuned models are then evaluated across five distinct math environments. Hyperparameter details are provided in Appendix F, with the training curve shown in Figure 5(b), and Pass@1 accuracy reported in Table 1.

The math grader used for reward and evaluation is based on HuggingFace's `math_verify` library[6]. We found that even minor differences in grading logic across codebases yields substantial variation in reported performance. Thus, all results should be interpreted comparatively—within a consistent evaluation framework—rather than as absolute values. This further highlights the need for unified benchmarking, as provided by GEM.

Results in Table 1 reveal a clear and consistent pattern: across all environments, performance improves substantially after RL training compared to the base model. Furthermore, the model with access to the Python tool achieves higher final performance in every setting.

Table 2: QA benchmark scores for the base agent and agents trained with different RL configurations. † and * denote single-hop and multi-hop datasets, respectively.

| Qwen3-4B | Base (no tool) | Base + RL (no tool, single env) | Base + RL (no tool, mixed env) | Base + RL (with tool, single env) | Base + RL (with tool, mixed env) |
|---|---|---|---|---|---|
| NQ† | 6.1 | 15.4 | 15.8 | 35.0 | 37.3 |
| TriviaQA† | 35.4 | 43.4 | 44.9 | 69.0 | 71.9 |
| PopQA† | 11.3 | 19.0 | 19.9 | 47.1 | 48.1 |
| HotpotQA* | 11.1 | 21.1 | 22.1 | 43.2 | 45.5 |
| 2wiki* | 10.0 | 26.8 | 30.1 | 44.5 | 46.7 |
| Musique* | 2.9 | 4.7 | 5.5 | 17.6 | 19.9 |
| Bamboogle* | 17.6 | 28.8 | 28.8 | 49.6 | 48.8 |
| Average | 10.2 | 22.7 | 23.9 | 43.7 | **45.5** |

---

[6]`github.com/huggingface/Math-Verify`

We also perform a parallel analysis for QA tasks, this time integrating the Search tool. We train on two environment compositions: `qa:HotpotQA` alone, and a mixture of both `qa:HotpotQA` and `qa:NaturalQuestions`. All other setting are the same as for the Math experiments (see above). Evaluation spans seven diverse QA environments. Results, summarized in Table 2, mirror those from the math domain: RL finetuning markedly improves performance, and models equipped with the Search tool achieve the highest accuracy in every scenario.

The consistency of these findings across both domains (mathematics and QA), tools (Python and Search), and multiple evaluation environments underscores the flexibility and robustness of GEM's approach to RL LLM with tool integration.

## 4.4 STUDYING GENERALIZATION

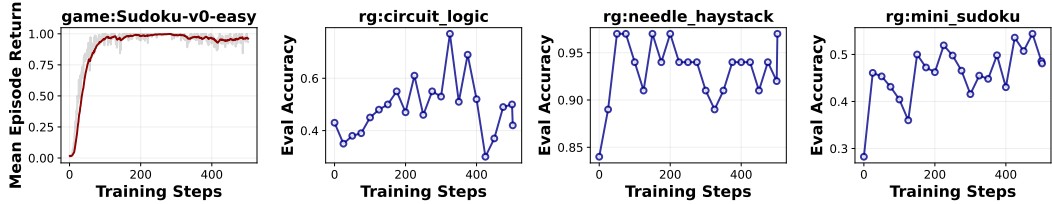

Figure 6: Training on the `game:sudoku-v0-easy` environment generalizes to ReasoningGym.

GEM's environments can be used for both training and evaluation. This makes it ideal for investigating cross-environment generalization. For instance, we demonstrate training on the `game:sudoku-v0-easy` environment, while periodically evaluating on three different environments, with some encouraging initial generalization results shown in Figure 6.

## 4.5 INTEGRATION WITH TRAINING FRAMEWORKS

Finally, we demonstrate that GEM—which takes care of the environment side—can be easily integrated with five popular frameworks that handle the training side. There has been a proliferation of frameworks focusing on the training side of LLM RL. These often rely heavily on multiple other libraries (such as vLLM for response generation (Kwon et al., 2023), and DeepSpeed for optimization (Rasley et al., 2020)). The diverse range of features and design choices make it challenging for researchers to select and adapt a suitable training framework to their specific needs.

To address this GEM comes with complete, single-file training scripts showing clean integration into five widely used LLM RL frameworks: Oat (Liu et al., 2024), Verl (Sheng et al., 2024), OpenRLHF (Hu et al., 2024), ROLL (Wang et al., 2025a), and RL2 (Tan et al., 2025). These are validated in Figure 7(a) where we show the training curve for each. Despite minor differences due to underlying design choices of the frameworks (e.g., different LLM generation engines) and RL stochasticity, all curves exhibit similar trends, demonstrating that GEM is agnostic to training frameworks and validating their implementation equivalence. Furthermore, supporting a wide range of frameworks allows us to effortlessly access their advanced features. For example, enabling the asynchronous rollout in RL2 gives an immediate 2× gain in wall-clock efficiency (Figure 7(b)).

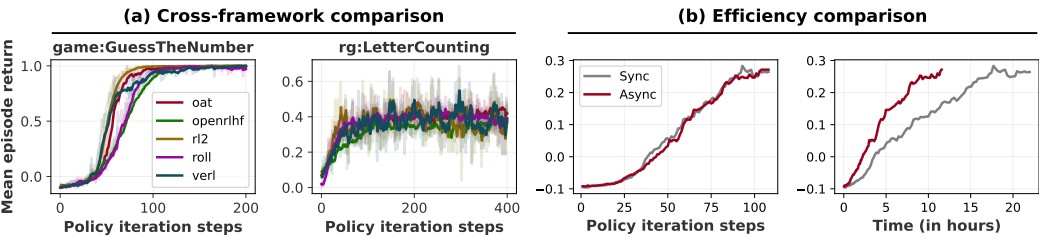

Figure 7: **(a)** Training curves on two environments showing successful integration of GEM into five existing frameworks. **(b)** Asynchronous rollout improves wall-clock efficiency of training Sudoku-solving agents based on Qwen3-4B-Base.

## 5 AGENT EVALUATION WITH GEM

In addition to RL training, GEM can serve as a **unified evaluation interface** to test LLM agents' performance. In this section, we present two example use cases where we evaluate agents powered by strong LLMs (GPT-5 (OpenAI, 2025), Gemini-2.5-Pro (Gemini Team, 2025) and Claude-Sonnet-4 (Anthropic, 2025a)) on two complex tasks: database operation via model context protocol (MCP) (Anthropic, 2025b) and terminal interaction via docker containers, both of which have been added to GEM following Appendix A.

### 5.1 GENERAL TOOL USE VIA MODEL CONTEXT PROTOCOL

Modern LLM agents often need to interact with external tools, such as search engines, APIs, and code interpreters. To facilitate this, GEM is designed to be compatible with the MCP, which is an open protocol that provides a standardized way for LLMs to communicate with external tools and data sources.

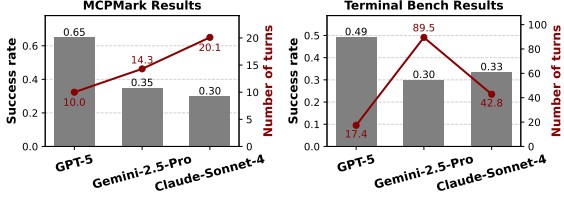

Figure 8: Benchmark results on MCPMark (Postgres subset) and Terminal-Bench (subset) using GEM as a unified evaluation toolkit.

The MCP architecture consists of an MCP host (the LLM application), an MCP client, and an MCP server (the external tool). By adopting this protocol, GEM allows for "plug-and-play" tool usage, where any tool that implements the MCP server interface can be used by an agent in a GEM environment without custom integration. This significantly simplifies the process of creating tool-augmented LLM agents and opens up a vast ecosystem of potential tools.

Using a PostgreSQL MCP tool, we assess the agent's tool-augmented reasoning capabilities using 20 database tasks taken from MCPMark (Team, 2025a). We report the average success rate and the average number of turns required to complete the tasks in the left panel of Figure 8[7]. GPT-5 attains the best success rate with the fewest interactions, while Gemini-2.5-Pro and Claude-Sonnet-4 have slightly lower and varied performance.

### 5.2 TERMINAL ENVIRONMENT VIA DOCKER CONTAINER

To support a wider range of tasks, especially those involving complex software dependencies and interactions with the operating system, GEM includes support for environments running inside docker containers. The integrated terminal environment provides a sandboxed unix operating system where agents can learn to perform tasks using shell commands. This approach provides a high degree of isolation and reproducibility, ensuring that the environment is consistent across different machines.

We assess the terminal mastery of LLM agents on 57 tasks sampled from Terminal-Bench (Team, 2025b), without any scaffolding. The right panel of Figure 8 reports the average success rate and the number of turns required to complete the tasks. GPT-5 attains the highest success rate with the most efficient interaction, followed by Claude-Sonnet-4 and Gemini-2.5-Pro. The evaluation leverages the same interaction loop used for RL training, highlighting GEM's role as a unified framework for both reinforcement learning and standardized evaluation.

## 6 CONCLUSIONS

GEM aims to accelerate agentic LLM research by providing a decoupled and clean library that is agnostic to training frameworks, a unified agent-environment interface and a suite of standardized environments. In this paper, we introduced the design choices of GEM, the current suite of task domains and tools, features like vectorized environment execution, a simple yet general multi-turn REINFORCE algorithm implemented in five training frameworks, a comprehensive algorithm benchmarking evaluation, and in-depth analysis on several algorithmic details. We invite the community to enter the era of experience for LLM agent learning, and join us in both using and continuing to develop the GEM framework.

---

[7]Our evaluation relies on the basic response generation API rather than agent frameworks (e.g., LangChain, OpenAI Agent SDK), which may lead to deviations from the original benchmark results.

ACKNOWLEDGMENT

This project was partially supported by the Singapore Ministry of Education Academic Research Fund Tier 1 (Award Number: T1 251RES2514) and the Ministry of Digital Development and Information (MDDI) under the Singapore Global AI Visiting Professorship Program (Award No. AIVP-2024-002).

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

## A   ENVIRONMENT REGISTRATION

GEM enables rapid development of new RL environments. In this section, we illustrate two scenarios: (i) integrating additional datasets into an existing task and (ii) defining a custom task, followed by the procedure for registering these environments for use.

The following code snippet shows how to add a new dataset for math environment, where the answer verification logic is predefined by GEM and can be reused.

```python
import gem
from gem.envs.registration import register

register(
    "math:GSM8K-Example",
    "gem.envs.math_env:MathEnv",
    dataset_name="axon-rl/GSM-8k", # HuggingFace or local dataset path
    question_key="problem",
    answer_key="answer",
)

env = gem.make("math:GSM8K-Example") # ready to use
```

Next, we demonstrate how to build a new environment from scratch by defining the initial state distribution (in `.reset()`) and the transition and reward functions (in `.step()`) as follows.

```python
from gem.core import Env
from gem.envs.registration import register
from gem.utils.constants import TERMINAL_STATE
from gem.utils.parsing import extract_last_boxed_answer

class ReverseStringEnv(Env):
    def __init__(self, str_len: int = 5):
        super().__init__()
        self.str_len = str_len

    def _get_instructions(self):
        return (
            "You are tasked to reverse a given string.\n"
            "You may provide your response in any manner. Only the content wrapped inside \\
                boxed{} will be considered as your final answer.\n"
            f"Please reverse the string: {self.gt_str}.\n"
        )

    def reset(self, seed=None):
        super().reset(seed)
        characters = string.ascii_letters + string.digits # A-Z, a-z, 0-9
        self.gt_str = "".join(random.choices(characters, k=self.str_len))
        return self._get_instructions(), {}

    def step(self, action):
        clean_action = extract_last_boxed_answer(action)
        if clean_action is None:
            reward = 0
        else:
            reward = float(clean_action[::-1] == self.gt_str)
        return TERMINAL_STATE, reward, True, True, {}

# Register your environment
register("custom:ReverseString", ReverseStringEnv)

env = gem.make("custom:ReverseString")
```

## B   CASE STUDIES OF LANGUAGE GAMES

This section illustrates example interactions of game-playing agents trained on `GuessTheNumber` under different discount factors $\gamma$. With a near-undiscounted setting ($\gamma = 0.999$), the agent fails to complete the game within the trial budget, as it lacks incentive to obtain the terminal correctness reward as earlier as possible (see example below).

($\gamma = 0.999$, the agent exhausts all 10 trials)

**State:** You are playing Guess The Number. You have to guess the number between 1 and 50 (inclusive) within 50 turns. $\cdots$ As you play, the history of your guesses will be appended below. Use the information to complete the game before you run out of guesses.
Enter your first guess to start the game.
At turn 1, you guessed 14, and the target number is higher than 14.
At turn 2, you guessed 22, and the target number is higher than 22.
$\ldots$
At turn 10, you guessed 39, which has been already guessed before.
Enter your next guess.

**Action:** Let's break down the problem step by step:
1. **Initial Guess (Turn 1):** The target number is higher than 14. This means the target number is between 15 and 50 (inclusive).
2. **Initial Guess (Turn 2):** The target number is higher than 22. This means the target number is between 23 and 50 (inclusive).
$\ldots$
From the final two steps, the target number is between 37 and 50 (inclusive). The final answer is \boxed{38}.

In contrast, when using a more myopic return estimate ($\gamma = 0.9$), the agent is encouraged to minimize the number of turns, discovers the binary search strategy, and converges to the optimal policy.

($\gamma = 0.9$, the agent learns to conduct binary search)

**State:** You are playing Guess The Number. You have to guess the number between 1 and 50 (inclusive) within 50 turns. $\cdots$ As you play, the history of your guesses will be appended below. Use the information to complete the game before you run out of guesses.
Enter your first guess to start the game.
At turn 1, you guessed 25, and the target number is lower than 25.
At turn 2, you guessed 12, and the target number is higher than 12.
At turn 3, you guessed 18, and the target number is higher than 18.
At turn 4, you guessed 21, and the target number is higher than 21.
At turn 5, you guessed 23, and the target number is lower than 23.
Enter your next guess.

**Action:** To determine the target number between 1 and 50, we can use a **binary search strategy**, which is effective for narrowing down a range when given hints like 'higher' or 'lower.'
Step-by-Step Reasoning:
1. **Initial Guess (Turn 1):** - The target number is lower than 25. - This means the target number is in the range [1, 24].
2. **Second Guess (Turn 2):** - The target number is higher than 12. - This means the target number is in the range [13, 24].
3. **Third Guess (Turn 3):** - The target number is higher than 18. - This means the target number is in the range [19, 24].
4. **Fourth Guess (Turn 4):** - The target number is higher than 21. - This means the target number is in the range [22, 24].
5. **Fifth Guess (Turn 5):** - The target number is lower than 23. - This means the target number is in the range [22, 22].
Final Guess:
The target number is in the range [22, 22], which means the target number is 22. **Final Answer:** \boxed{22}

## C  ALGORITHM

For completeness we include the full algorithm of ReBN in Algorithm 1.

---

**Algorithm 1** Multi-turn REINFORCE with Return Batch Normalization (ReBN)

---

**Require:** Policy $\pi_\theta$, Environment $\mathcal{M} = (\mathcal{S}, \mathcal{A}, P, R, \gamma, \rho)$, Batch size $B$
1: **while** not converged **do**
2:    Reset batch buffer $\mathcal{B} \leftarrow \emptyset$
3:    **while** $|\mathcal{B}| \leq B$ **do**
4:        // **Multi-turn episode collection**
5:        Sample the initial state $s_0 \sim \rho$
6:        **for** turn $t = 0, 1, \ldots, T - 1$ until terminate **do**
7:            $y_t \sim \pi_\theta(\cdot | s_t)$                          ▷ Generate reasoning + action
8:            $a_t \leftarrow \text{extract\_action}(y_t)$
9:            $r_t \leftarrow R(s_t, a_t)$
10:           $s_{t+1} \leftarrow P(s_t, a_t)$
11:       **end for**
12:       **for** $t = 0, 1, \ldots, T - 1$ **do**
13:           $G_t \leftarrow \sum_{k=t}^{T-1} \gamma^{k-t} r_k$                  ▷ Compute discounted return
14:           Add $(s_t, y_t, G_t)$ to $\mathcal{B}$
15:       **end for**
16:    **end while**

17:    // **Return Batch Normalization**
18:    $\tilde{G}_i \leftarrow (G_i - \text{mean}(\mathbf{G})) \, / \, \text{std}(\mathbf{G})$

19:    // **Policy optimization**                        ▷ Or proximal update for data reuse
20:    Update $\theta$ using Monte Carlo policy gradient $\sum_{i=1}^{B} \tilde{G}_i \nabla_\theta \log \pi_\theta(y_i | s_i)$
21: **end while**

---

## D  EXTENDED EMPIRICAL STUDIES WITH GEM

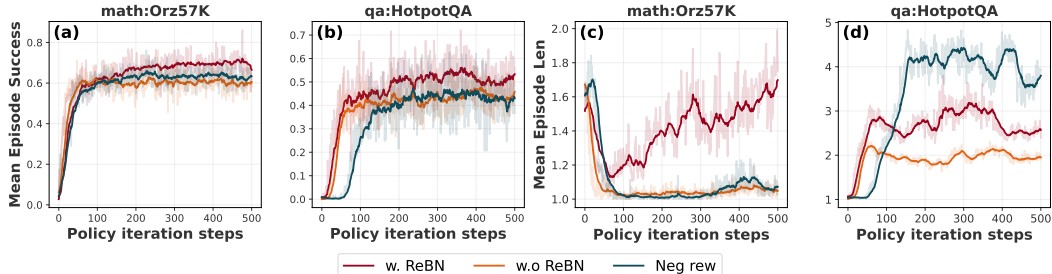

Figure 9: Learning curves of different reward shaping strategies. (**a-b**) The average success rate of two environments. (**c-d**) The corresponding average number of turns taken to solve the tasks, equal to the number of tool calls minus one.

### D.1  IMPROVING LEARNING EFFICIENCY VIA RETURN BATCH NORMALIZATION (REBN)

As briefly discussed in Section 4.1, while REINFORCE demonstrates strong performance across most environments, its convergence can be suboptimal in certain cases. To investigate this further, we present an in-depth ablation study here. Following minimalist principles, we began with the vanilla REINFORCE algorithm and a simple reward scheme: $r = 1$ for correct answers and $r = 0$ otherwise. This approach has been shown effective for single-turn RL training (Singh et al., 2023; Xiong et al., 2025). However, as shown in Figure 9(c) (w.o ReBN), it failed to induce tool usage in multi-turn settings, despite significant amount of initial attempts.

We hypothesize that this failure arises from the absence of *negative gradients* under 0/1 reward shaping, which are crucial for efficient learning and exploration. To address this, we introduced negative gradients in two ways: (i) assigning fixed negative rewards ($r = 1$ for correct and $r = -1$ for incorrect answers, denoted as Neg rew in Figure 9); and (ii) applying Return Batch Normalization with

0/1 rewards, where Monte Carlo returns in REINFORCE are normalized as described in Algorithm 1 (denoted as ReBN in Figure 9). While both 0/1 and $\pm 1$ reward schemes theoretically induce the same optimal policy, they might exhibit markedly different learning dynamics in practice.

Notably, ReBN demonstrates strong and consistent performance across environments—not only in math and QA tasks (Figure 9) but also in all other settings (Figure 1). We also observe that models can be sensitive to fixed reward shaping: for example, Neg rew fails to improve tool use in `math:Orz57K`, yet leads to tool overuse in `qa:HotpotQA`, both of which are suboptimal behaviors.

## D.2 RL ON VISION-LANGUAGE ENVIRONMENTS

In addition to text-only environments, we support visual elements as part of the observation for the agent to understand and take actions. As a demonstrative example, we build a visual-language environment based on Geometry3k dataset[8] for training reasoning agent to solve geometry math problems with images input. We RL-tune Qwen2.5-VL-3B/7B-Instruct (Bai et al., 2025) using Dr. GRPO (Liu et al., 2025b), and the learning curves are shown in Figure 10. An example reasoning trace is shown in Figure 11.

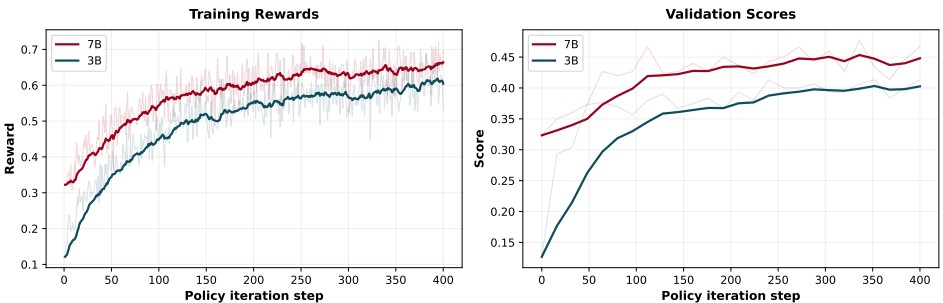

Figure 10: Learning curves of vision-language agents. We RL-tune Qwen2.5-VL-3B/7B-Instruct using Dr. GRPO on the `math:Geometry3K` environment and track their training rewards (left) and validation scores (right).

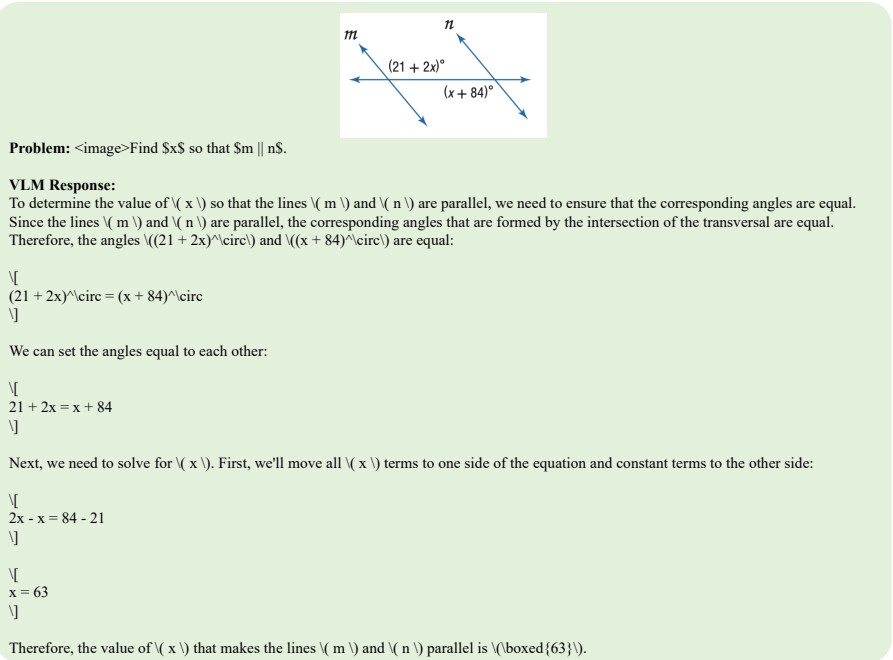

Figure 11: An example problem and the response of a trained agent based on Qwen2.5-VL-7B-Instruct.

---

[8]https://huggingface.co/datasets/hiyouga/geometry3k.

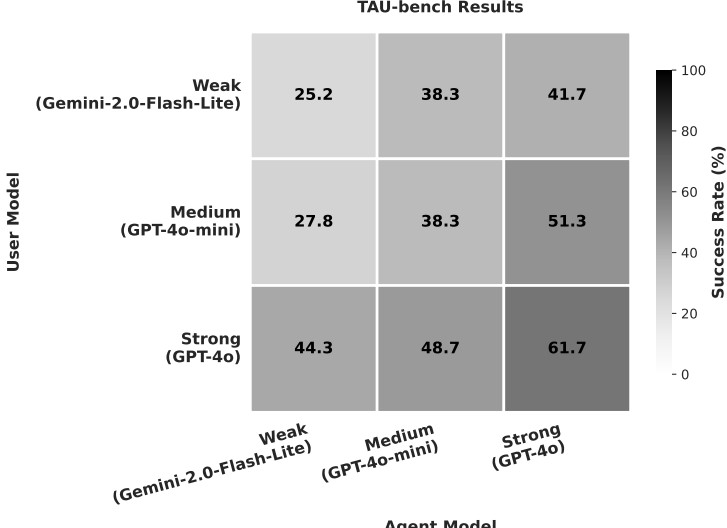

Figure 12: Multi-agent evaluation on TAU-bench retail. Stronger user simulators (rows) consistently improve agent performance (columns) across model strengths.

## D.3 MULTI-AGENT ENVIRONMENTS

Many scenarios considered within reinforcement learning, such as adversarial games, social dilemmae and multi-party control systems require multiple agents by definition (Albrecht et al., 2024). This introduces a whole host of different design decisions which a general multi-agent API should be able to embrace. By studying previous approaches to defining multi-agent APIs (Terry et al., 2021; Lanctot et al., 2019; Bou et al., 2023; Hu et al., 2023) and condensing them into a simple abstraction scheme, GEM also provides a high-level API for multi-agent LLM RL.

In doing so, GEM enables convenient development of agents that can collaborate, compete, or simulate realistic interactions with other agents/entities, adding comprehensiveness to its extensible platform.

**Design.** GEM's multi-agent API is designed with the following principles in mind:

- **Minimal Prescriptions.** The API only imposes constraints and methods when it is helpful to do so. This includes the `AgentSelector` class which allows for arbitrary definitions of turn order via overriding.
- **One Abstraction, One API.** There should be only one way to realize the environment abstraction regardless of the environment's properties.

**API overview.** GEM provides a `MultiAgentEnv` base class that extends the standard Gym API to support multiple agents. The `step()` and `reset()` functions operate on dictionaries keyed by agent identifiers:

```python
from gem.envs.multiagent import MultiAgentEnv

env = MyMultiAgentEnv()
observations, infos = env.reset() # Dict[agent_id, observation]

while not done:
    actions = {agent_id: agent.act(obs) for agent_id, obs in observations.items()}
    observations, rewards, terminations, truncations, infos = env.step(actions)
    done = all(terminations.values())
```

To implement a custom environment, users inherit from `MultiAgentEnv` and implement `observe(agent)` and `_process_actions(actions)`. The framework handles agent lifecycle management and cumulative rewards tracking. Turn coordination is managed via `AgentSelector`, which supports two modes: *sequential* (agents act one at a time in round-robin order) and *parallel* (all agents act simultaneously). The environment can then determine

which agents are active at each step and automatically advance turns, enabling flexible multi-agent interaction patterns without manual bookkeeping.

**TAU-bench retail integration.** We demonstrate this API by integrating the TAU-bench retail benchmark (Yao et al., 2024), which evaluates conversational agents on customer service tasks. We formulate this as a two-agent environment: an `assistant` agent using tools (order lookup, product search) and a `user` agent simulating customer behavior via an LLM. The user simulator is initialized with task instructions and generates queries; the assistant must satisfy these requests before episode termination.

**Impact of user model strength.** A key question in multi-agent RL is: *how does simulated user agent capability affect trainable assistant agent learning?* We vary both user and assistant models across three levels: weak (Gemini-2.0-Flash-Lite), medium (GPT-4o-mini), and strong (GPT-4o), yielding 9 configurations to study user-assistant model interactions.

Evaluating across all 115 tasks from the TAU-bench retail test set (Figure 12), we find that stronger user agents consistently improve overall success rates across all assistant agent model strengths. Notably, the strongest assistant (GPT-4o) exhibits the largest absolute performance gains (20% from weak to strong user), achieving 61.7% success with a strong user simulator. Interestingly, a strong user paired with a weak assistant (44.3%) outperforms a weak user paired with a strong assistant (41.7%), demonstrating that improving the *user agent* is crucial for robust conversational task completion. These results motivate us to develop multi-agent RL to co-evolve user and assistant agents to achieve scalable and autonomous learning.

# E    RELATED WORKS

There is a significant body of work on tool-integrated language models—including SkyRL-v0 (Cao et al., 2025), VerlTool (Jiang et al., 2025), ReCall and ReSearch (Chen et al., 2025a), Search-R1 (Jin et al., 2025), ReTool (Feng et al., 2025a), and SimpleTIR (Xue et al., 2025). A common design pattern in these methods is to collect multi-turn agent-environment interactions as single continuous sequences of tokens of agent actions interleaved with environment outputs. Training then simply involves masking the environment outputs from the loss calculation.

However, this single-sequence approach presents two significant limitations. First, the state observation is rigidly defined as the complete history of actions and outputs. This restricts the ability to use alternative state representations, such as pruning "thinking" tokens or summarizing the history to avoid exceeding context lengths. Second, this formulation inherently limits the reward structure to a single, trajectory-level signal, preventing the use of finer-grained, per-step rewards, and effectively fixing the discount factor at $\gamma = 1$. In Section 4.2 we demonstrate that $\gamma < 1$ is crucial for obtaining the optimal fastest search behavior. By contrast, with trajectory-level rewards, the natural speed incentive from $\gamma < 1$ is lost, and hence other works, such as SimpleTIR, must tune and enforce a strict turn-limit to get this behavior.

To address this, our framework, GEM, is designed for maximum flexibility by collecting trajectories as a sequence of individual transitions (i.e., state, action, reward, next state) as in the full, unsimplified RL formulation. This design choice enables arbitrary state observation constructions (using observation wrappers), and also preserves compatibility with per-turn rewards and arbitrary discount factors $\gamma \leq 1$. The verl-agent framework (Feng et al., 2025b) also adopts this transition-wise approach, which enables its implementation of GiGPO (Feng et al., 2025b), an RL method that utilizes turn-level rewards. While GiGPO collapses to trajectory-level GRPO when observations are unique, it is an example of a type of algorithm that is now straightforward to implement with GEM's infrastructure.

There are multiple popular frameworks that focus on the agent training side (e.g., Oat (Liu et al., 2024), Verl (Sheng et al., 2024), OpenRLHF (Hu et al., 2024), ROLL (Wang et al., 2025a), and RL2 (Tan et al., 2025)). Currently, many works that build on these, including verl-agent, RAGEN (Wang et al., 2025b), Verlog (Chen et al., 2025b), and many of the works above, add environments by directly modifying the source code. This results in tight coupling between training and environments, and makes it difficult to maintain and reuse the environments for future research. As a result, each codebase tends to support only a small, ad-hoc collection of environments, making it hard to compare different methods. Even environments with the same name are often inconsistent between codebases. GEM addresses this by dealing with all the environment infrastructure, including providing

a diverse suite of environments, and corresponding baselines. This makes it easy to keep training and environments decoupled, with the aim of freeing researchers from cumbersome environment development and setup, and thus enabling quicker prototyping and evaluation of new ideas.

We also note that there are early works (Abdulhai et al., 2023; Tajwar et al., 2025) incorporating text games as the evaluation toolkit or for language model fine-tuning. However, they did not focus on a standardized suite of RL training environments nor did they investigate different RL algorithms.

## F  EXPERIMENTAL SETTINGS

All our experiments are performed on $8 \times$ A100 GPUs and finished in about one day. The detailed experimental configurations are shown in Table 3.

Table 3: Hyperparameter configurations used in all experiments.

| Parameter | Value |
|---|---|
| ACTOR | |
| Maximum response length per turn | 4096 tokens |
| Sampling temperature, train | 1.0 |
| Sampling temperature, evaluation | 0.0 |
| (top P, top k) | (1.0, -1) |
| LEARNER | |
| Optimizer | AdamW |
| Adam parameters $(\beta_1, \beta_2)$ | (0.9, 0.95) |
| Weight decay | 0.0 |
| Gradient norm clipping | 1.0 |
| Learning rate scheduler | Constant |
| Learning rate | $1 \times 10^{-6}$ |
| Inner proximal update epoch | 2 |
| KL loss coefficient | 0.0 |
| KL penalty coefficient | 0.0 |
| Policy clipping parameter | 0.2 |
| Discount factor | 0.9 (**game**,**qa**); 1.0 (otherwise) |
| GAE $\lambda$ | 0.95 |
| Steps | 500 |

