# OpenReview forum: "GEM: A Gym for Generalist LLMs"
_ICLR.cc/2026/Conference — ICLR 2026 Poster_

### Official Review · Reviewer_rtWh · 2025-10-30

**Soundness:** 3
**Presentation:** 3
**Contribution:** 3
**Rating:** 6
**Confidence:** 4

**Summary:**

The paper presents GEM, which is a unified and reproducible environment suite for reinforcement learning with LLMs. GEM offers an OpenAI/Gym style API with asynchronous vectorization, tool integration (Python, Search, MCP), and multiple task types (Math, Code, QA, Reasoning, Terminal).
The authors benchmark REINFORCE, PPO, and GRPO, showing their Return Batch Normalization (ReBN) variant improves stability and efficiency across multi-turn tasks. GEM integrates with major RL frameworks and provides evaluation suites like Terminal-Bench and MCPMark. The main strengths are its clean environment API and comprehensive benchmarking; the main drawbacks are no released code and limited technical comparison to existing APIs.

**Strengths:**

- I love the well-structured, reproducible environment interface modeled after OpenAI Gym. Asynchronous vectorized execution is cool, but I would advise against auto-reset which in my experience provides a very marginal gain and introduces a lot of pain when dealing with last observations of a trajectory. The clear separation between environments and trainers fills a major reproducibility gap left by prior frameworks like Verl and OpenRLHF.
- Enabling turn-level rewards and arbitrary γ is a nontrivial practical and conceptual upgrade, although other frameworks like torchrl already provide this.
- The paper has good benchmarks and ablations. The major frameworks are compared which gives a good intuition of the strengths of the library.

**Weaknesses:**

- not being able to see the codebase makes me worried about accepting the paper. Reproducibility is the base of any scientific advance, and since the core claim is about standardization and benchmarking, lack of release undercuts the entire point.
- The related work acknowledges VerlTool, Verl-Agent (GiGPO), OpenRLHF, and RL2, but only superficially. Since the paper is about a technical advance and a new standardization of environment APIs, a clear comparison should have be undertaken. It doesn’t discuss TorchRL’s EnvBase or OpenEnv at all — which now also propose standardized step/reset interfaces and multi-process async collection.
- It's unclear if GEM scales beyond small local setups or integrates well with cluster-level runners like Ray / Monarch or others.
- ReBN shouldn't be framed as a new algorithm but as a useful implementation tweak.

**Questions:**

- Why not releasing the code in an anonymized way? That would help to judge how easy (or not) it would be for the community to adopt it (as well as doc, testing etc)
- Is the vision to make GEM a parallel ecosystem of environments, or build extension points and integrations with other frameworks (recent OpenEnv or extension for VeRL etc)? The ecosystem of RL post-training libraries is already quite crowded, so interoperability with similar frameworks is key.
- Can the authors provide throughput or scaling benchmarks (e.g., vectorized rollout speed, GPU utilization) to substantiate the claim that GEM enables high-throughput training beyond local setups?
- Does GEM does use Ray or other RPC actor services, or any distributed rollout pool? Distributing actors across a cluster is key in settings where the tool / simulator requires a lot of compute resources.

---

> ### Author Response · Authors · 2025-11-21
>
> **W1,Q1. Code availability**: We apologize for not releasing anonymized code with our submission. We have now done this here: https://anonymous.4open.science/r/anonymous_gem-142F
>
> **W2. Comparison detail to related works**: We thank you for asking about this. GEM is the most comprehensive and flexible environment framework of existing works. We have shown a more detailed comparison in the table below. We will add this comparison to the related works section (as well as moving the related works section into the main paper).
>
>
> | | Type of environments | Framework dependency | Extra features | Tool integration | Time of official release |
> | :--- | :--- | :--- | :--- | :--- | :--- |
> | TorchRL.EnvBase | Gym, math | PyTorch | Vectorization | Python (added recently) | 2023.06 |
> | VerlAgent | QA, Games | VeRL | Parallel env, visual input | search | 2025.05 |
> | VerlTool | Mostly QA with different tools | VeRL | Async rollout, visual input | Search, python, terminal, SQL | 2025.09 |
> | OpenEnv | Games, code, QA, browser | Agnostic | Isolated execution | Python, browser | 2025.10 |
> | **GEM** | Games, code, QA, webshop, math, terminal-bench | Agnostic | Parallel env, async execution, visual input | Search, python, terminal | ICLR submission in 2025.09 |
>
>
> **W3. Scalability to cluster-level runners**: We would like to clarify that GEM is designed as an unified and easy-to-use environment executor, instead of a framework or a system. It is therefore compatible with any cluster-level training framework (e.g., VeRL which is based on Ray), and can thus be scaled arbitrarily. Thank you for asking this. We will make this point clearer in the paper.
>
> **W4. Position of ReBN**: We completely agree that ReBN is a “useful implementation tweak” rather than a novel new algorithm. In our text, we tried to make this clear, for example, in line 74 we stated “this paper also presents and discusses a simple yet effective algorithmic variant of REINFORCE which incorporates Return Batch Normalization (ReBN)”. We will adjust the wording in the paper to make this clearer.
>
> **Q2. Purpose**: Our anticipated role for GEM is as a standardized, parallel ecosystem of environments, instead of extensions on other frameworks. Hence, unlike VeRL-Tool or VeRL-Agent which is built upon VeRL, GEM is a standalone library and agnostic to training frameworks. In Section 4.5, we showed it’s fairly easy to integrate GEM with any other RL LLM training codebases, validating its interoperability. The integration codes can be found in the /examples folder.
>
> **Q.3 Substantiate high-throughput**: In Figure 7b, we show the wall-clock efficiency improvements after enabling async rollout during training multi-horizon Sudoku agents. We can observe that the overall training throughput is nearly doubled while maintaining the same learning curves. We will add similar run-time and GPU utilization plots for other environments for completeness.
>
> **Q4. Distributed setup**: Thanks for this great question. Instead of using Ray of RPC services, GEM directly starts the environment in the local process. We chose this design to enable compatibility with existing training frameworks, since we can then host GEM as services (e.g., with RESTful APIs) to be called by the training frameworks’ distributed rollout engines/actors. Our training framework integration scripts show several examples of this.
>
> —
>
> We thank the reviewer for their careful and detailed review of our submission which will help us significantly improve the paper. Please let us know if you have any remaining concerns. If not, we humbly ask if you would consider raising your score.

---

### Official Review · Reviewer_LQDp · 2025-10-31

**Soundness:** 3
**Presentation:** 4
**Contribution:** 3
**Rating:** 8
**Confidence:** 4

**Summary:**

This paper introduces GEM, a open-source framework designed to facilitate the training and evaluation of agentic RL (LLM-based). Drawing an analogy to OpenAI-Gym's role in traditional RL, GEM provides a standardized environment-agent interface, a diverse suite of multi-turn environments, and features for efficient experimentation like asynchronous vectorized execution. The paper also proposes a simple yet effective baseline algorithm, REINFORCE with ReBN, which is shown to be well-suited for multi-turn settings with dense rewards.

**Strengths:**

Overall, this paper addresses a critical need for standardized infrastructure of agentic RL. It provides a foundational tool that can accelerate progress and improve the rigor of research in this area.

1. Comprehensiveness of the GEM Framework: GEM with a clean API, support for asynchronous execution , a diverse suite of tasks (games, reasoning, code, QA), modular tool integration (Python, Search), and extensibility through wrappers. This represents a significant  engineering effort.
2. Rigorous and Insightful Empirical Evaluation: The paper includes a thorough benchmarking of several key RL algorithms.
3. Practicality and Usability: The authors have clearly prioritized usability by providing ready-to-use, single-file integration scripts for five popular LLM RL training frameworks. This significantly lowers the adoption barrier and demonstrates that GEM is agnostic to the training infrastructure.

**Weaknesses:**

Novelty of ReBN: The core algorithmic proposal, ReBN, is an effective application of a well-known technique (return/advantage normalization) in RL. The paper is transparent about this, but it's worth noting that the primary novelty of the paper lies in the framework (GEM) itself, rather than in a breakthrough RL algorithm.

**Questions:**

ReBN vs. Advantage Normalization: Could you please elaborate on the empirical or theoretical trade-offs between using ReBN without a critic versus using a learned critic with GAE and advantage normalization? Is the primary benefit of ReBN its implementation simplicity and reduced computational overhead?

Roadmap: This is an excellent starting point for a valuable community resource. Could you share more about your future vision for GEM? Specifically:
1. Do you plan to incorporate more open-ended, sandbox-style environments that involve multi-turn interaction with persistent state and feedback?
2. Have you considered structuring the benchmarks into different difficulty tiers or with specific focuses to allow for more targeted evaluation?
3. Regarding framework integration, beyond individual scripts, have you considered designing a more universal and simple interface or adapter to further lower the cost of connecting to new and existing RL frameworks?

---

> ### Author Response · Authors · 2025-11-21
>
> We thank you for the strong positive score and for highlighting the "comprehensiveness" and "usability" of GEM.
>
> **W1. Novelty of ReBN**: We are grateful that you appreciated the purpose of GEM in that “the primary novelty of the paper lies in the framework (GEM) itself, rather than in a breakthrough RL algorithm.” We elaborate and emphasize this in our General Response 3.
>
> **Q1. ReBN vs. critic-based advantage normalization**: You are exactly right in that the primary benefits of ReBN over critic-based normalization (e.g. PPO) is the simplicity and reduced cost by not needing a critic model. Querying the critic model is both memory and compute intensive, and empirically value models are difficult to train [1]. ReBN removes the critic baseline and instead uses batch statistics for normalization. We present ReBN as a simple baseline algorithm that maintains the critic-free simplicity of GRPO whilst being compatible with the full multi-turn RL problem.
>
> **Q2. Open-ended environments**: Thank you for asking this. We have already integrated MCP (Model Context Protocol) (Section 5.1), which allows GEM agents to connect to external servers (Notion, GitHub, Databases). This moves GEM from a "simulator" to a real-world agent interface. Our next plan is to develop a fully open-ended deep research task, for example inspired by the work in [2].
>
> **Q3. Difficulty tiers**: Thank you for noting this. We do have a standardized set of different levels registered in the codebase. For example, we have “sudoku-easy”, “sudoku-hard”, etc. We will add the details of this into the revision.
>
> **Q4. More universal and simple interface**: The current structure is the most standard and universal interface we could think of. We therefore do not currently have plans for this, but if you have any ideas of how to make it more universal we would love to discuss as we are always striving to improve GEM.
>
> [1] - DeepSeekMath: Pushing the Limits of Mathematical Reasoning in Open Language Models. Z Shao, et al. 2024. https://arxiv.org/abs/2402.03300.
>
> [2] - DeepResearcher: Scaling Deep Research via Reinforcement Learning in Real-world Environments. Y Zheng, et al. 2025. https://arxiv.org/abs/2504.03160.
>
> —
>
> We thank the reviewer for their encouraging review and detailed feedback. Please let us know if you have any more questions or comments as we would be happy to discuss further.

---

### Official Review · Reviewer_hvWT · 2025-10-31

**Soundness:** 3
**Presentation:** 3
**Contribution:** 3
**Rating:** 6
**Confidence:** 4

**Summary:**

This paper introduces GEM (General Experience Maker), an open-source environment simulator for agentic RL. Apart from a standardized framework for the environment-agent interface, GEM features a diverse suite of environments, robust integrated tools, and single-file example scripts demonstrating using GEM with five popular RL training frameworks. Additionally, authors also propose REINFORCE with Return Batch Normalization (ReBN), designed for multi-turn agentic LLM tasks.

**Strengths:**

1. I appreciate the work authors did for building such an integrated environment simulator, which I believe will facilitate future research on agentic LLM.

**Weaknesses:**

1. The proposed REINFORCE with ReBN, seems like a trivial extention of GRPO to multi-turn tasks. Further practical investivation and theoretical foundation are required to fully justify its effectiveness.
2. While the unification of existing benchmarks is a valuable contribution, this work would have been even more solid and appealing had the authors also analyzed the deficiencies of current benchmarks and built new ones based on that analysis.

**Questions:**

Please refer to Weaknesses above.

---

> ### Author Response · Authors · 2025-11-21
>
> We appreciate your support for the unified simulator and your recognition of the "robust integrated tools."
>
> **W1. Triviality of ReBN**:
> We agree that ReBN is a simple change from GRPO or REINFORCE. Its purpose in GEM, however, is not as a novel new algorithm, but instead as a simple multi-turn baseline against which to test future algorithms. One reason why the surprisingly strong empirical results (particularly on `game` environments) are interesting is that, theoretically ReBN does introduce some bias due to normalizing by a quantity that depends on past actions. (Theoretically this bias reduces as the batch size tends to infinity.) Additionally the division by the standard deviation means that ReBN optimizes a proxy objective, similar to GRPO. Empirically we find that ReBN works well despite this. We initially did not consider detailed analysis of ReBN to be within the scope of the paper, but we agree the theory is interesting and will include detail of the theory in the updated version of the paper. Thank you for asking about this and helping us improve on the work.
>
> **W2. Analysis of current benchmarks**:
> Thank you, this is a really interesting question and goes back to our original motivation for this project. We were finding that, without any standardized environment package, different codebases each had their own implementations of environments. For example, even for the simplest single-turn math RL environment, the differences in reward functions can lead to discrepancies in the final scores – *Please see our discussion in **General Response 2***.
>
> Additionally, while our game environments are based on TextArena, we found its original sparse reward design is insufficient for effective multi-turn agent learning. We thus adapted them significantly to improve reward shaping, making these environments solvable while still challenging, hence useful for algorithmic benchmarks.
>
> —
>
> We thank the reviewer again for their careful review of our submission. We hope we have sufficiently addressed all of your questions. Please let us know if you have any remaining concerns. If we have sufficiently addressed them all we humbly ask if you would consider raising your score.

---

### Official Review · Reviewer_5brH · 2025-11-12

**Soundness:** 3
**Presentation:** 3
**Contribution:** 2
**Rating:** 2
**Confidence:** 4

**Summary:**

The paper introduces an RL environment interface for LLM agents supporting multi-step (phrased as multi-turn in the paper) tool use. It aggregates tasks including ReasoningGym, text-based games (TextArena), coding, math, and QA, and integrates tools such as Python interpreters, search engines, and the MCP. The authors also benchmark a reinforce with Return Batch Normalization (ReBN) baseline across these environments.

**Strengths:**

- Engineering quality: The paper/report is well-written and represents a solid engineering effort, particularly in offering a unified interface compatible with five major training frameworks (Oat, Verl, OpenRLHF, ROLL, RL2).
- Clarity: The motivation for transitioning from single-turn to multi-turn agentic workflows is clearly presented.

**Weaknesses:**

In general, I believe this is a solid engineering work which is helpful to many practitioners, however, the novelty of both the environment and the algorithmic design may limit its contribution to the academia.
- Limited novelty (environments): The contribution is largely an aggregation of existing tasks and datasets (e.g., GSM8K, TextArena, ReasoningGym) rather than a fundamental/novel research contribution in environment design. The differentiation from prior tool-integrated frameworks is minimal, and the scale of environments and experiments are limited.
- Limited novelty/relevance (algorithm): The proposed ReBN algorithm is presented as a contribution, but normalizing returns/advantages is a standard technique in the RL literature and does not constitute an original algorithmic contribution. It may be more meaningful to propose algorithms that focus on the special challenges of multi-step tool use scenarios in the real world.
- Please place related works in the main body of the paper instead of downweighting them in the appendix. It is important to properly give credits to prior works.

**Questions:**

- Would it be possible to include open-ended environment design with more practical real-world usages in the current framework? If so, If so, what specific engineering/coding improvements would be necessary?
- Training for long-horizon tool use suffers from many specific issues like temporal credit assignment. The proposed ReBN only serves as generic regularization, could the authors discuss more principled algorithmic designs specifically tailored to address more specific challenges inherent in agentic RL?
I am happy to raise my scores if the concerns raised can be sufficiently addressed.

---

> ### Author Response · Authors · 2025-11-21
>
> We thank you for acknowledging the “solid engineering effort” and clarity of our work. We aim to address your concerns below:
>
> **W1. Novelty of environments and infrastructure**: To clarify, the aim of GEM is not to provide new tasks or infrastructure, but rather to encourage the community to coalesce to using the same set of standardized environments. Currently each codebase includes its own implementations of environments, and we’ve observed seemingly identical environments can produce very different results across codebases due to subtle, hidden implementation differences (please see General Response 2 and Table 1 for details).
>
> GEM addresses this challenge by providing a robust framework with all the necessary components for environment infrastructure (including a standardized set of complex diverse environments, parallelization and asyncronicity, fully benchmarked baselines, comprehensively tested example integration scripts with five common training frameworks, etc). Our hope is for GEM to become the default environment framework, enabling fair, consistent, and reproducible comparisons of new algorithms. Additionally note that GEM is not limited by scale and is fully compatible with large-scale multi-node training frameworks.
>
> **W2. Novelty of algorithm**: We emphasize that we present ReBN, not as a novel algorithm, but as a simple, surprisingly effective baseline for comparing future algorithms to. A secondary motivation of GEM is to facilitate the progression from single- to multi-turn environments. We therefore arrived at ReBN as it is compatible with the full multi-turn RL problem while also maintaining GRPO’s critic-free simplicity. We fully benchmark ReBN (alongside GRPO and PPO), not to demonstrate superior performance, but instead to show they can act as strong and well-understood baselines for evaluating new algorithms. (See General Response 3 for additional discussion.)
>
> **W3. Related works in the main paper**: Thank you for this suggestion. We completely agree with this and will move the related works section to the main paper (especially with the extra page allowance for the camera-ready version).
>
> **Q1. Open-ended environment design**: We demonstrate this in Section 5 with the Terminal (Docker container) and MCP (Model Context Protocol) environments. This includes the infrastructure for sandboxing, async execution of external tools, etc. These allow agents to interact with real operating systems and databases, which are inherently open-ended. Beyond this we plan to add an open-ended “deep research” task, building on the existing MCP infrastructure.
>
> **Q2. Algorithmic designs for long-horizon tasks**: As described in response to W2, we feel that development of novel algorithms is beyond the scope of our paper. The aim with GEM is to provide all the necessary infrastructure to facilitate future work into developing new principled multi-turn algorithms. Note we also include PPO–an existing principled multi-turn algorithm–as a comprehensively benchmarked baseline.
>
> —
>
> We thank the reviewer for their encouraging feedback. We hope we have been able to address your concerns about contribution. Reviewer LQDp raised many of the same comments about the role of GEM and novelty of ReBN and gave a Score=8, noting that, GEM “provides a foundational tool that can accelerate progress and improve the rigor of research in this area,” and that “it's worth noting that the primary novelty of the paper lies in the framework (GEM) itself, rather than in a breakthrough RL algorithm.“ Please let us know if you have any more questions or concerns as we would be more than happy to discuss further. If we have been able to address all your concerns we humbly ask if you would consider raising your score.

---

### Author Response · Authors · 2025-11-21
**General Response**

We thank the reviewers for their thoughtful feedback and for recognizing GEM as a "solid engineering effort" (Reviewer 5brH), "well-structured" (Reviewer rtWh), and a framework that "addresses a critical need" (Reviewer LQDp) with "excellent" presentation (Reviewer LQDp).

We would like to address three common questions:

**1. Code availability**: It was amiss of us not to anonymize and release the code. We have now uploaded it: https://anonymous.4open.science/r/anonymous_gem-142F/README.md. Thank you for prompting us to do this. We are committed to continually improving GEM and so welcome any feedback.

**2. Novelty and contribution**: Reviewer 5brH questions novelty. We would like to emphasize that the primary contribution of GEM is as a standardized, diverse set of environments with robust, easy-to-use infrastructure, not the invention of new tasks or algorithms. The reason we feel this is needed is that currently even environments with the same name vary widely between codebases. For example, in the table below we show the “performance” of the same model on the same task (MATH) when evaluated using reward functions from different codebases.



**Table 1**: *We find that evaluating the same responses in different codebase’s implementations of the same environment gives results for the same set of responses. Here we take the MATH training set/environment and generate 8 responses per question for 2000 randomly sampled questions using the DeepSeek-R1-Distill-Qwen-1.5B with 8K token budget. We then grade the same responses using the reward function from different open-source codebases. The differences in performance are similar in size to the differences often presented between different algorithms, thus environment implementation is currently a significant confounder when comparing the performance of algorithms.*

| [math\_verify](https://github.com/huggingface/Math-Verify) | [prime\_math](https://github.com/volcengine/verl/blob/main/verl/utils/reward_score/prime_math/__init__.py) | [mathruler](https://github.com/hiyouga/MathRuler) | [oat\_eval](https://github.com/sail-sg/understand-r1-zero/blob/main/understand_r1_zero/math_grader.py) |
| :--- | :--- | :--- | :--- |
| 46.71 | 52.95 | 54.19 | 53.06 |




The reason for the difference shown in the table is the differences in the string parsing and grading logic of the reward functions. Similar hidden discrepancies exist across many different tasks, which makes it extremely difficult to fairly evaluate and compare algorithms and models. Our hope is that, by building infrastructure with a standardized set of environments, hopefully future works will use GEM for apple-to-apple fair comparison. To support this we designed GEM’s decoupled structure so that it is easy to integrate with different codebases (which we demonstrate in Figure 7a where we demonstrate single-file integration with 5 different training frameworks).
We thank the reviewers for raising this. We will use this feedback to help us improve the paper by adding a case study with these results to explain our initial motivations and to make GEM’s purpose in the community clearer.




**3. The Role of ReBN**: A secondary motivation for GEM is to facilitate and accelerate the move from single-step environments to complex multi-step environments. As part of this we wanted to include a baseline that maintains GRPO’s critic-free simplicity, but, unlike GRPO, is compatible with the full multi-turn problem (turn-level rewards and discount factors $\gamma < 1$). We therefore arrived at ReBN, which we present, not as a breakthrough novel algorithm, but as a sensible and surprisingly strong baseline which future works can easily be built on. We will make the purpose of presenting ReBN clearer in the revision.

—

We thank all the reviewers for their invaluable feedback, and we answer their individual questions below.

---

### Author Response · Authors · 2025-12-01
**Final comment by authors**

Dear Area Chair,

Since the scores were reverted due to the accident happened earlier, we hope to take the opportunity to note on the previous score improvement by reviewer 5brH, who **raised the score from 2 to 6 after our rebuttals**.

With that, we believe our rebuttals have addressed the major concerns and led to overall positive scores. We hope you could take into consideration when assessing our paper.

Thank you for your time and efforts!

Best,\
The Authors

---

### Meta-Review · Area_Chair_E6MK · 2026-01-07

**Summary:**

Reviewers universally recognized GEM as a solid, valuable engineering effort that addresses a standardization gap for LLM agent research. The primary debate centered on the paper's novelty and the framing of the ReBN algorithm. Concerns were also raised about code availability (resolved) and detailed comparisons with related work.

**Reviewer Concerns:**

The authors' rebuttal successfully addressed the major practical concern by releasing the code. They also clarified that GEM's main contribution is the standardized framework itself, and ReBN is presented as a simple baseline, not a novel algorithm. This addressed core concerns of Reviewers 5brH and LQDp. Outstanding concerns include the need for a more detailed technical comparison with existing environment libraries (rtWh) and a deeper discussion of the theoretical properties/limitations of ReBN (hvWT).

**Reviewer Scores:**

5brH would maintain their raised score of 6.

hvWT might marginally increase to a 7 if satisfied with promised additions on ReBN theory.

rtWh might increase to a 7 given code release and promised related work expansion.

LQDp would likely maintain their strong 8.

---

### Decision · Program_Chairs · 2026-01-26

Accept (Poster)